# Dual-Input Isolated DC-DC Converter with Ultra-High Step-Up Ability Based on Sheppard Taylor Circuit

**Chih-Lung Shen \***, **Li-Zhong Chen and Hong-Yu Chen**

Department of Electronic Engineering, National Kaohsiung University of Science and Technology, Kaohsiung City 82445, Taiwan; 0652807@nkust.edu.tw (L.-Z.C.); 0352030@nkust.edu.tw (H.-Y.C.)
**\*** Correspondence: clshen@nkust.edu.tw; Tel.: +886-925-871-685

**Abstract:** A dual-input high step-up isolated converter (DHSIC) is proposed in this paper, which incorporates Sheppard Taylor circuit into power stage design so as to step up voltage gain. In addition, the main circuit adopts boosting capacitors and switched capacitors, based on which the converter voltage gain can further be improved significantly. Since the proposed converter possesses an inherently ultra-high step-up feature, it is capable of processing low input voltages. The DHSIC also has the important features of leakage energy recycling, switch voltage clamping, and continuous input-current obtaining. These characteristics advantage converter efficiency and benefit the DHSIC for high power applications. The structure of the proposed converter is concise. That is, it can lower cost and simplifies control approach. The operation principle and theoretical derivation of the proposed converter are discussed thoroughly in this paper. Simulations and hardware implementation are carried out to verify the correctness of theoretical analysis and to validate feasibility of the converter as well.

**Keywords:** dual-input converter; high voltage gain; leakage energy recycling; galvanic isolation; voltage clamping

## 1. Introduction

Nowadays, electricity is mostly generated by fossil fuels and nuclear fuel. Although nuclear power plants can generate considerable power by utilizing a little amount of nuclear fuel, nuclear waste influencing the environment is inevitable. Fossil fuels have been overused and become in shortage. Therefore, human beings attempt to discover more alternatives for maintaining global economic development and environment protection. To alleviate the problems of global temperature rising and serious greenhouse gas emission, many kinds of clean-energy power generation, such as photovoltaic (PV) panel, wind turbine, and fuel-cell stack, are developed imperatively [1,2].

In general, renewable power systems need a DC-bus voltage in the range of 380 V–400 V for grid-tied connection or in high power applications. Unfortunately, the output voltage of a PV module, battery or fuel cell is much lower than the dc-bus voltage and thus conventional DC/DC converters cannot be utilized directly to serve as an energy interface for dealing with power control. In addition, if the power supplied from two input ports has to be processed simultaneously, dual-input converter with high voltage gain is essentially anticipated. Figure 1 illustrates a hybrid generation system that includes two sources, a high step-up converter, and an inverter for AC application, which reveals that a dual-input converter with high step-up voltage gain is urgent in such system.

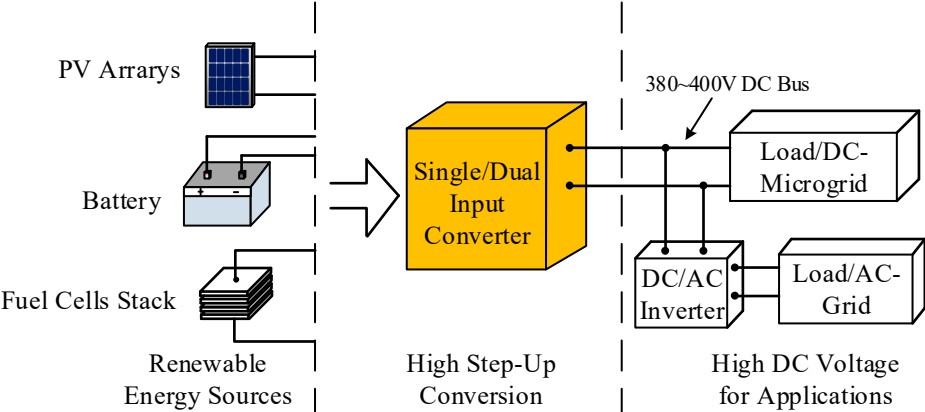

**Figure 1.** An illustration of the hybrid renewable-energy generation system.

Theoretically, conventional step-up converters, like Boost and Flyback [3–6], can achieve a high voltage gain under the operating with extremely high duty ratio or the design of a much higher turns ratio. However, extreme high duty cycle or turns ratio will dramatically degrade the conversion efficiency due to large conduction loss and copper loss of windings. In order to improve conversion efficiency and voltage gain, many transformerless high step-up converters are proposed [7–10]. Although these converters are designed to achieve high step-up characteristics with reasonable duty ratio, here still exits some problems, for instance, large transient current and limited voltage gain, confining converter flexibility. To mitigate the mentioned drawbacks, converters incorporating coupled inductor and/or switched capacitor are proposed [11–14], however, which are only able to process single-input source. In order to deal with two different kinds of inputs, dual-input converters (DIC) are proposed. In comparison with single input, a dual generation system is capable of providing higher reliability, durability and power rating. The structure of DICs can be simply classified as series type and parallel one. The conventional series-type DICs construct string connection at input ports [15,16]. Such a DIC will malfunction in case that either of the two inputs fails. The parallel-type DICs collocate different sources in parallel so that even if one of the inputs is out of commitment, it still can accomplish voltage stepping to meet DC-bus level [17–23]. Some of them are controlled with time-sharing scheme. That is, only one DC source is permitted to delivery its energy to the load at a time. Compared with series-type DICs, parallel ones possess much better features from the aspects of reliability and controllability. Nevertheless, limitation on voltage gain is unavoidable, which confines converter applications in the field of high DC-bus voltage requirement.

In order to convert a lower DC voltage to a much higher level and to provide consecutive power even under the situation that one input source shuts down, this paper proposes a DHSIC, which is developed by means of boosting capacitor, switched capacitor and Sheppard Taylor circuit. The proposed dual-input converter can achieve the following important features: ultra-high step-up ability, continuous input current, and galvanic isolation. Furthermore, the proposed converter possesses the competence of inherent voltage clamping without any additional devices and recycling leakage energy stored in transformers. Therefore, the voltage spike on the power switch can be suppressed and converter efficiency is also improved.

The structure of this paper is organized as follows. Following the introduction, the operation principle of the proposed dual-input high step-up isolated converter is described in Section 2. The steady-state analysis is discussed in Section 3, which covers voltage gain of the converter, voltage and current stresses of the semiconductor device, and magnetizing inductance design in continuous conduction mode (CCM). To verify the correctness of proposed converter, experimental results from a 200-W prototype are illustrated in Section 4. Finally, Section 5 summarizes the conclusions of this paper.

## 2. Operation Principle of Proposed Dual-Input Converter

The equivalent of the proposed DHSIC is shown in Figure 2. Parameters in Figure 2 are represented in the following. $V_{in1}$ and $V_{in2}$ are input voltages, while $i_{in1}$ and $i_{in2}$ denote input currents. The practical model of coupled inductor includes magnetizing inductance, leakage inductance, and an ideal transformer. The $L_{m1}$ and $L_{m2}$ are the magnetizing inductances of $T_1$ and $T_2$, respectively, meanwhile, leakage inductances are expressed as $L_{k1}$ and $L_{k2}$. The $S_1$–$S_4$ represent the four main power switches. The $C_1$ and $C_2$ function as boosting capacitors, and $C_3$ and $C_4$ serve as switched capacitors. These capacitors can elevate converter voltage gain effectively. The $D_1$–$D_6$ are rectifier diodes. In addition, $D_o$ and $C_o$ are the output diode and filter capacitor, respectively. Output voltage and current of DHSIC are in turn described as $V_o$ and $I_o$. Finally, the output equivalent resistance is presented as $R_o$. Even though the proposed DHSIC works normally in dual input operation (DIO), it still possesses the ability to be in single-input operation (SIO) while either input source fails. In Section 2, DIO will be first discussed followed by SIO.

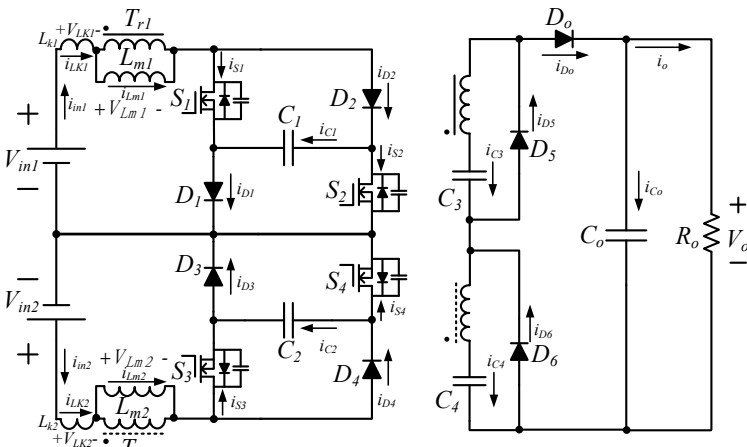

**Figure 2.** Equivalent circuit of the proposed converter.

### 2.1. Dual-Input Operation

The switches $S_1$ and $S_2$ are turned on/off simultaneously, so do the switches $S_3$ and $S_4$. Assume that the turn-on period of $S_1$ and $S_2$ is $D_1 T_s$ and $D_2 T_s$ is for $S_3$ and $S_4$. In addition, the magnitude of $V_{in2}$ is twice that of $V_{in1}$. While the proposed DHSIC operates at DIO and in continuous conduction mode (CCM), converter operation over one switching cycle can be divided into six states. Converter operation will be described state by state as the following proceeds with. In addition, the equivalent of each state and conceptual waveforms are depicted in Figures 3 and 4, respectively.

- State 1 [$t_0$~$t_1$]: Converter operation begins at this state, in which all switches $S_1$–$S_4$ are turned on at $t = t_0$. All diodes are reverse except the output diode $D_o$. The currents of leakage inductances, $i_{Lk1}$ and $i_{Lk2}$, increase linearly and steeply. Meanwhile, the energy stored in magnetizing inductances $L_{m1}$ and $L_{m2}$ is released to the output via transformers and diode $D_o$. When leakage inductance current rises to be equal to magnetizing current, the diode current flowing through $D_o$ will drop to zero and then this state ends. The diode $D_o$ turns OFF under zero current transition. That is, the reverse-recovery problem at $D_o$ can be therefore overcome.

- State 2 [$t_1$~$t_2$]: In State 2, the switches $S_1$–$S_4$ remain ON. The diodes $D_1$–$D_4$ and $D_o$ are reversely biased, but diodes $D_5$ and $D_6$ are forwarded. In this time interval, leakage inductance and magnetizing inductance of the coupled inductor $T_1$ absorb energy from $V_{in1}$ and $C_1$, similarly, $L_{k2}$ and $L_{m2}$ of $T_2$ from $V_{in1}$ and $C_1$. The voltage across $T_{r1}$ is $V_{in1} + V_{C1}$ and $T_{r2}$ is $V_{in2} + V_{C2}$. At the secondary of the DHSIC, switched capacitors $C_3$ and $C_4$ are charged by the energy from coupled

inductors $T_{r1}$ and $T_{r2}$, respectively. This state lasts for a time interval much longer than that of State 2 and is a major state in the converter operation.

- State 3 [$t_2$~$t_3$]: During the period from $t_2$ to $t_3$, switches $S_1$ and $S_2$ continue conducting. On the contrary, $S_3$ and $S_4$ are turned off at $t_2$. The diodes $D_1$–$D_4$ and $D_o$ are reversely biased, but $D_5$ and $D_6$ are in a forward state. The parasitic capacitances of $S_3$ and $S_4$ are charged and current $i_{Lk2}$ decreases dramatically. As the increasing voltages blocked by $S_3$ and $S_4$ reach $V_{C2}$, diodes $D_3$ and $D_4$ become forwarded and then the operation state enters State 4.

- State 4 [$t_3$~$t_4$]: All active switches remain the same on-off conditions as in State 3. That is, $S_1$ and $S_2$ are closed but $S_3$ and $S_4$ open. The voltage $V_{in1} + V_{C1}$ will supply $T_{r1}$ and forwards the energy to charge switched capacitor $C_3$. Meanwhile, the capacitor $C_2$ absorbs energy from $V_{in2}$ and $T_{r2}$. Leakage energy of $L_{k2}$ is recycled to $C_2$. The voltage stress of $S_3$ and $S_4$ will be clamped to $V_{C2}$. This operation state ends at the time both switches $S_1$ and $S_2$ are turned off.

- State 5 [$t_4$~$t_5$]: After $S_1$ and $S_2$ are turned off, the voltage across both switches increases. At the same time, their parasitic capacitances are charging toward the value of $V_{C1}$. With respect to the other switches, $S_3$ and $S_4$ are still in the OFF state. Once parasitic capacitance-voltage approaches to $V_{C1}$, State 5 ends and blocking voltage of $S_1$ and $S_2$ will be clamped at $V_{C1}$. The switched capacitor $C_3$ is still charging. During the time interval of State 5, the current flowing through $L_{k1}$ drops steeply. The diodes $D_1$ and $D_2$ will be forwarded at $t = t_5$ and then this state ends.

- State 6 [$t_5$~$t_6$]: From $t_5$ to $t_6$, all switches remain OFF. The $L_{m1}$ pumps its stored energy to charge $C_1$ and to the output as well. With respect to $L_{m2}$, it is also in energy-releasing but charges toward $C_2$, meanwhile, part of its energy will be transformed to the secondary of $T_{r2}$ to power the output. The leakage energy stored in $T_{r1}$ and $T_{r2}$ will be recycled to capacitors $C_1$ and $C_2$, respectively. During State 6, the series voltage of $T_{r1}$, $C_3$, $T_{r2}$, and $C_4$ is connected to the output. That is, the output can accordingly obtain a high voltage level. Like State 2 and 4, State 6 also plays a major role in the converter operation. While switches $S_1$–$S_4$ are turned on again at $t = t_6$, this state ends and converter operation over one switching cycle is completed.

## 2.2. Single-Input Operation

Once one of the inputs fails to supply power, the proposed converter still can function as a high step-up feature. Suppose that only $V_{in1}$ powers the DHSIC and in CCM condition. The converter will have four operation states over switching cycle. The corresponding key waveforms and equivalent circuits are illustrated in Figures 5 and 6 in turn.

- State 1 [$t_0$~$t_1$]: If only $V_{in1}$ supplies the converter, power processing is controlled by switches $S_1$ and $S_2$. Both switches are turned on at $t = t_0$ and converter operation begins. As shown in Figure 6a, the voltage across $T_{r1}$ will be equal to the series voltage of $C_1$ and $V_{in1}$. Magnetizing inductance $L_{m1}$ pumps its stored energy to the secondary of $T_{r1}$. Therefore, the current flowing through $L_{m1}$ decreases. The current $i_{Lk1}$ increases quickly. When $i_{Lk1}$ is equal to $i_{Lm1}$, this state ends. At this time, the current flowing through the diodes $D_o$ and $D_6$ also drops to zero. That is, the reverse-recovery problem of both diodes is therefore resolved.

- State 2 [$t_1$~$t_2$]: The equivalent of State 2 is illustrated in Figure 6b, in which the switches $S_1$ and $S_2$ remain closed. The $L_{m1}$ absorbs energy form $V_{in1}$ and $C_1$ and thus $i_{Lm1}$ increases linearly. At the secondary of $T_{r1}$, the switched capacitor $C_3$ is charging continuously over this stage. State 2 is a major state in the converter operation at SIO. At time $t = t_2$, the switches $S_1$ and $S_2$ are turned off and then the converter operation enters the next stage.

- State 3 [$t_2$~$t_3$]: In State 2, all diodes are in reverse bias except $D_5$. The parasitic capacitance of power switches $S_1$ and $S_2$ are charged and the current $i_{Lm1}$ decreases. The blocking voltage of $S_1$ and $S_2$ is therefore increasing. The associated equivalent is shown in Figure 6c. When the voltage across $S_1$ and $S_2$ approaches to $V_{C1}$, diodes $D_1$ and $D_2$ become forwarded. Then, State 4 starts.

- State 4 [$t_3$~$t_4$]: The equivalent circuit refers to Figure 6d, in which the magnetizing inductance forwards its stored energy to charge capacitor $C_1$ and to the output via the ideal transformer. Meanwhile, the leakage energy of $L_{k1}$ is recycled to $C_1$, which also suppresses the voltage spike on the active switches. Over the time interval from $t_3$ to $t_4$, switches $S_1$ and $S_2$ are open. With respect to diode, the $D_5$ is in reverse state but $D_1$, $D_2$, $D_6$ and $D_o$ are forward biased. State 4 is also a major state similar to State 2, dominating the converter operation. Both switches $S_1$ and $S_2$ will be turned on again at $t = t_4$ and then this state ends. Converter operation over one switching cycle is completed.

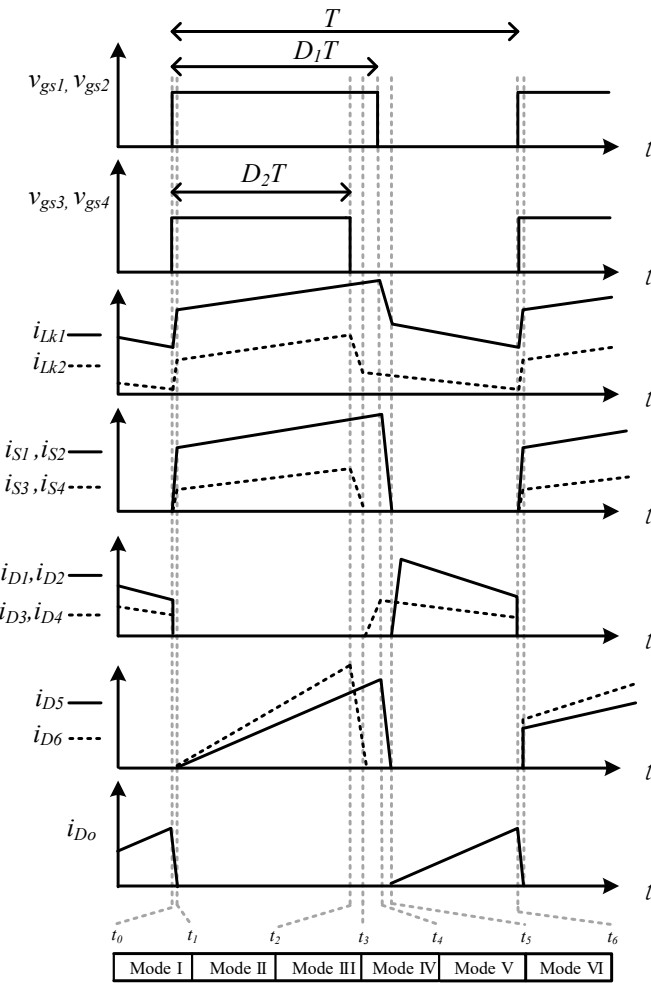

**Figure 3.** The conceptual key waveform of the proposed converter at DIO (dual input operation) and CCM (continuous conduction mode) operation.

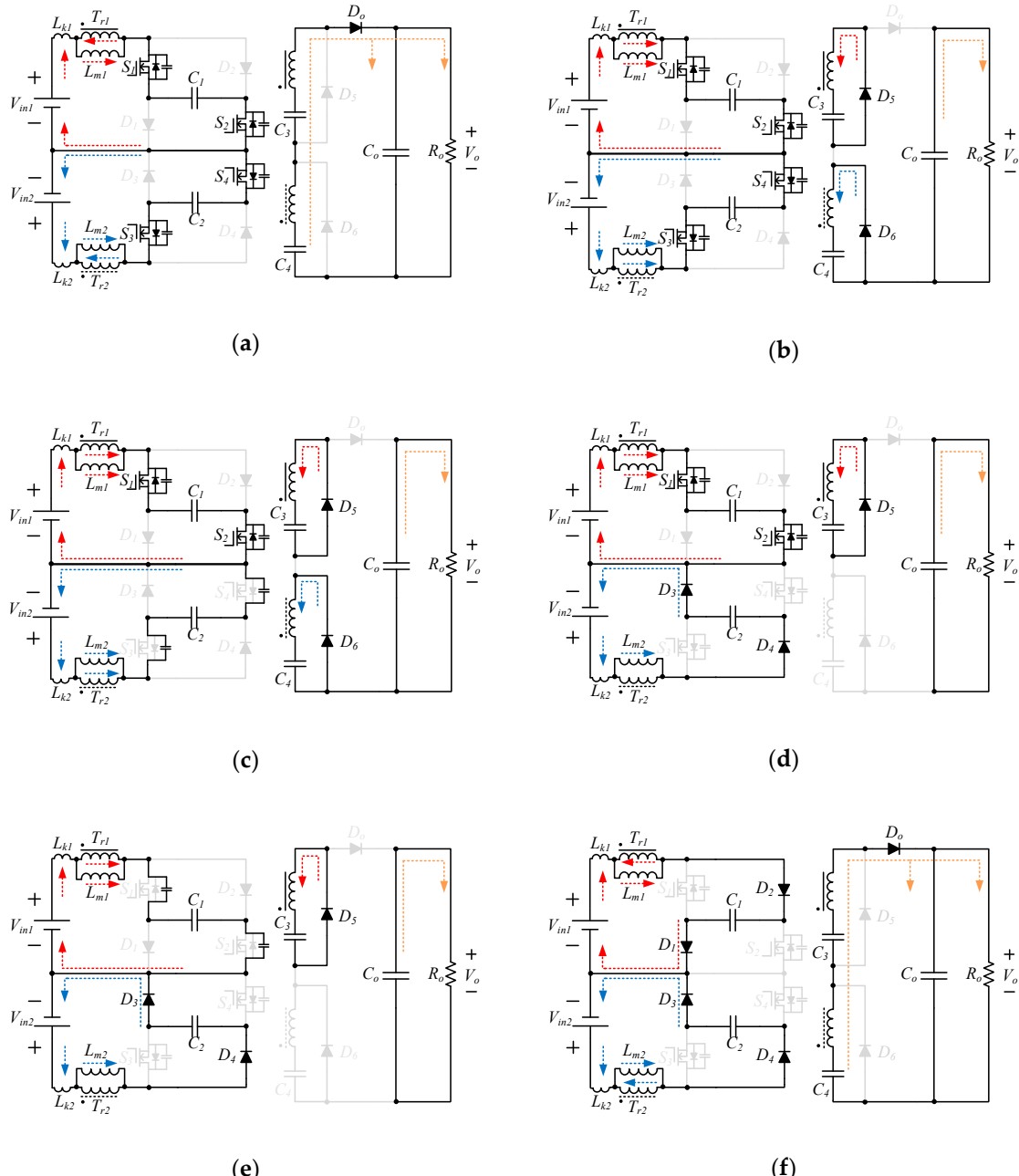

**Figure 4.** The equivalents of the proposed converter at DIO and CCM operation: (**a**) State 1, (**b**) State 2, (**c**) State 3, (**d**) State 4, (**e**) State 5, (**f**) State 6.

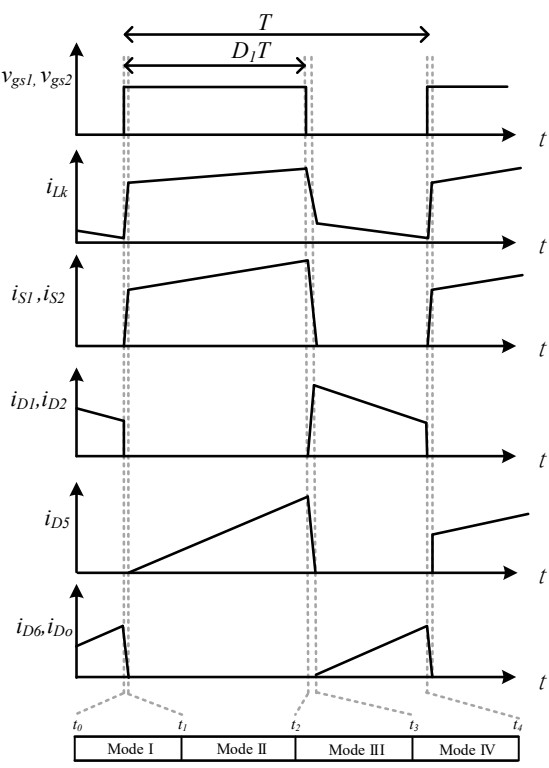

**Figure 5.** Conceptual key waveform of the proposed converter at SIO (single-input operation) and CCM operation.

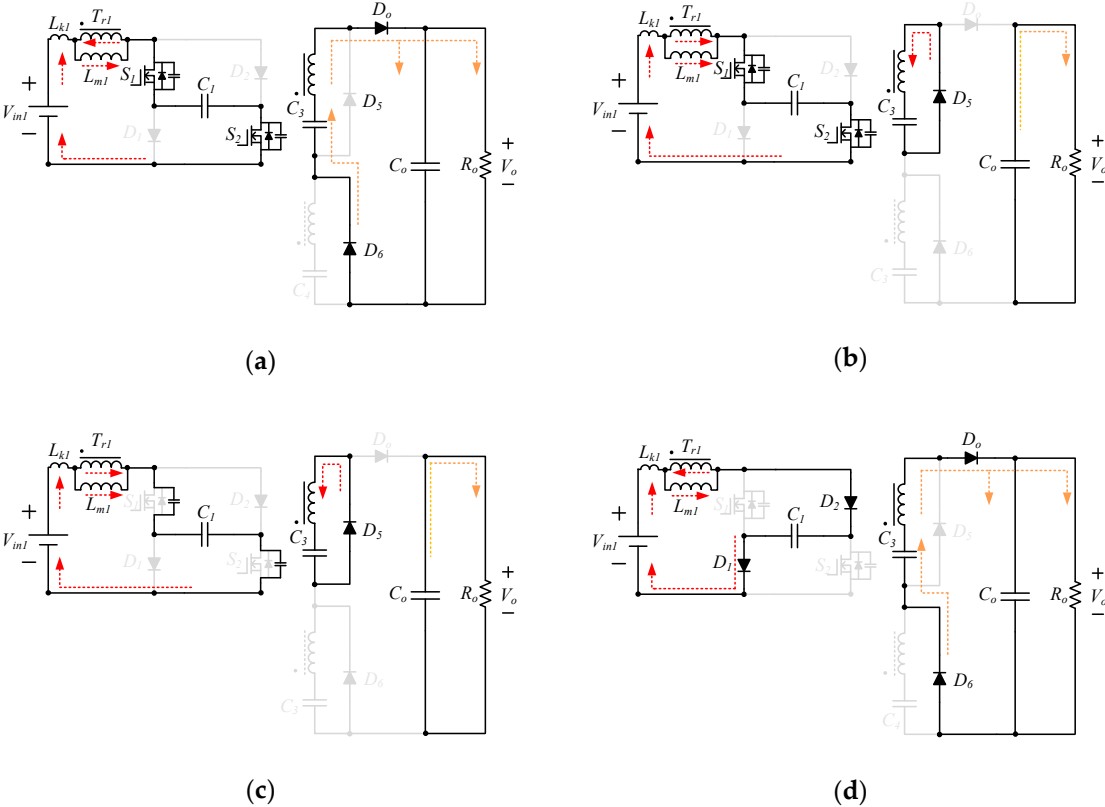

**Figure 6.** The equivalents of the proposed converter at SIO and CCM operation: (**a**) State 1, (**b**) State 2, (**c**) State 3, (**d**) State 4.

### 3. Steady-State Analysis of Proposed Converter

The voltage ratio of output to input, voltage stress and current stress of the semiconductor device, and magnetizing inductance design will be covered in this section. To simplify the steady-state analysis, the following assumptions are made.

1.  The values of the boosting capacitors $C_1$ and $C_2$ are large enough so as to keep their across voltages invariant.
2.  All diodes are regarded to be ideal. That is, forward drop voltage and ON-state resistance are neglected.
3.  The magnetizing inductance of the coupled inductor is much more than leakage inductance so that influence of the leakage inductance can be ignored.
4.  The turns ratios of the coupled inductors, $N_{1s}/N_{1p}$ and $N_{2s}/N_{2p}$, are defined as $n_1$ and $n_2$, respectively.
5.  The DHSIC is at CCM operation.

The driving pattern relating to the four switches is the same as that discussed in the previous section. The $S_1$ and $S_2$ are closed for $D_1T$ and $S_3$ and $S_4$ for $D_2T$. In addition, the duty ratio of $D_1$ is greater than $D_2$. Based on the assumptions made at the beginning of this section, states 2, 4 and 6 in Figure 4 will dominate the converter operation.

#### 3.1. Voltage Conversion Ratio

The proposed DHSIC can be regarded as a combination of two step-up converters which are symmetrical to common ground at the input side and in series connection at the output port. Furthermore, the two step-up converters are capable of operating individually. Therefore, the obtaining of voltage conversion ratio of the DHSIC can simply be derived from a single input situation and then to augment to dual-input situation. Suppose that only the $V_{in1}$ powers the DIC and both switches $S_1$ and $S_2$ are closed for $D_1T$ and open for $(1 - D_1)T$ over one switching period. The equivalents of switch ON and OFF are depicted in Figure 7. Applying voltage second balance criterion to $L_{m1}$ can yield

$$V_{Lm1,on} \cdot D_1T + V_{Lm2,off} \cdot (1 - D_1)T = 0 \tag{1}$$

From Figure 7b, the $V_{Lm1,on}$ and $V_{Lm1,off}$ can be obtained as follows:

$$V_{Lm1,on} = V_{in1} + V_{C1} \tag{2}$$

and

$$V_{Lm1,off} = V_{in1} - V_{C1}. \tag{3}$$

Substituting Equations (2) and (3) into Equation (1) can find the expression of $V_{C1}$:

$$V_{C1} = \frac{1}{1 - 2D_1} V_{in1}. \tag{4}$$

From Figure 7a, the voltage across capacitor $C_3$, $V_{C3}$, is found by the multiplication of turns ratio $n_1$ and $V_{Lm1,on}$, and then, from Equations (2) and (4) the $V_{C3}$ can be written as

$$V_{C3} = \frac{2n_1(1 - D_1)}{1 - 2D_1} V_{in1}. \tag{5}$$

According to Figure 7b, the following relationship holds:

$$n_1 V_{Lm1,off} = V_{C3} - V_o. \tag{6}$$

Based on Equations (3)–(6), the output voltage of the converter at SIO can be given by

$$V_o = \frac{2n_1}{1 - 2D_1} V_{in1}.$$

(7)

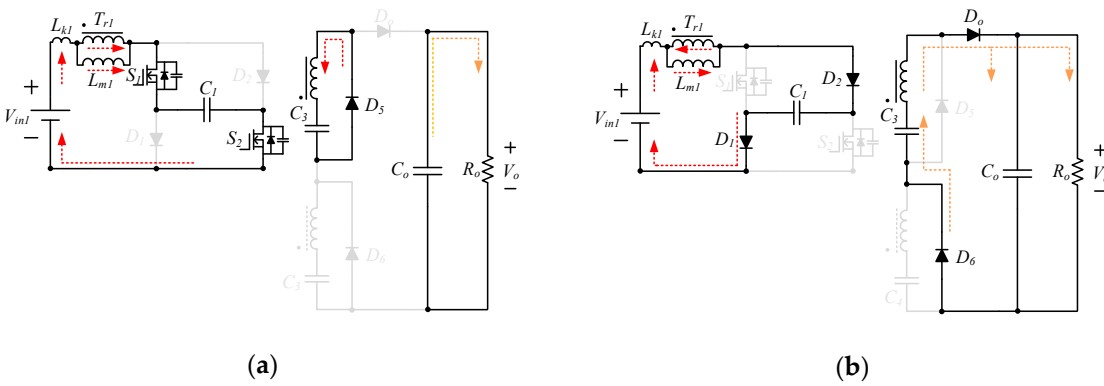

(**a**)                                                        (**b**)

**Figure 7.** The simplified equivalents while DHSIC (dual-input high step-up isolated converter) operates at SIO: during (**a**) switch-ON interval, (**b**) switch-OFF interval.

According to Equation (7), the switch duty ratio has to be less than 0.5, which is the converter limitation. Figure 8 depicts the relationship of voltage gain and duty ratio under different turns ratio.

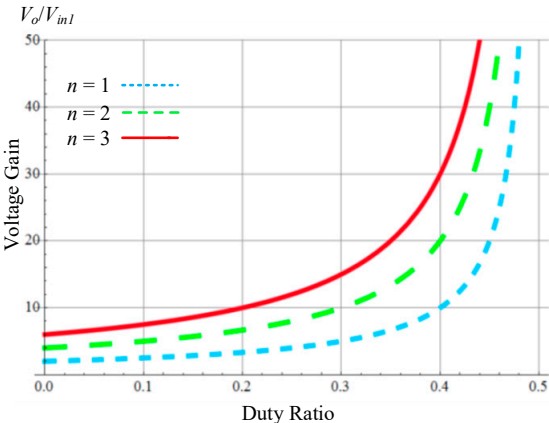

**Figure 8.** The voltage gain versus duty ratio under the SIO situation.

With respect to the DIO situation, the DHSIC has the same control scheme shown in Figure 3. The $S_1$ and $S_2$ are closed for $D_1T$ and open for $(1 - D_1)T$, while $S_3$ and $S_4$ are closed for $D_2T$ and open for $(1-D_2)T$ over one switching period. Equations (4) and (5) can be applied to the finding for the voltages of $C_2$ and $C_4$. That is, $V_{C2}$ and $V_{C4}$ are calculated by

$$V_{C2} = \frac{1}{1 - 2D_2} V_{in2}$$

(8)

and

$$V_{C4} = \frac{2n_2(1 - D_2)}{1 - 2D_2} V_{in2},$$

(9)

respectively. Being similar to Equation (6), the following relationship will hold under a dual-input situation.

$$n_1 V_{Lm1,off} + n_2 V_{Lm2,off} = V_{C3} + V_{C4} - V_o.$$

(10)

Thus, the corresponding output voltage at DIO of the converter can be described as

$$V_o = \frac{2n_1}{1-2D_1}V_{in1} + \frac{2n_2}{1-2D_2}V_{in2}. \tag{11}$$

Assume that $V_{in2} = 2V_{in1}$, $D_1 = D_2 = D$, and $n_1 = n_2 = n$, according to Equation (11), Figure 9 represents the relationship of voltage gain versus turns ratio, while under DIO situation.

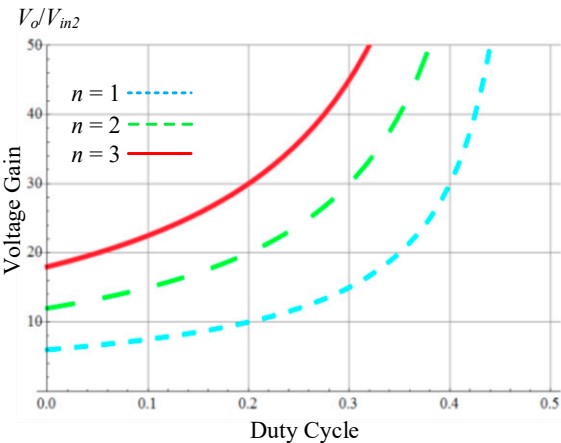

**Figure 9.** The voltage gain versus duty ratio under the DIO situation.

### 3.2. Voltage Stress of Semiconductor Device

According to the structure of the DHSIC, the semiconductor devices, $S_1$, $S_2$, $D_1$, and $D_2$, will have the same voltage stress. In addition, the power stage has inherently symmetrical configuration and is able to operate independently and individually at primary side. Therefore, the voltage stress of $S_1$, $S_2$, $D_1$, and $D_2$ can be determined from Figure 7, accordingly all of which will be equal to $V_{C1}$. From Equation (4), this voltage stress is expressed as

$$V_{S1,stress} = V_{S2,stress} = V_{D1,stress} = V_{D2,stress} = V_{C1} = \frac{1}{1-2D_1}V_{in1}. \tag{12}$$

Similarly, voltage stress of $S_3$, $S_4$, $D_3$, and $D_4$ can be determined by

$$V_{S3,stress} = V_{S4,stress} = V_{D3,stress} = V_{D4,stress} = V_{C2} = \frac{1}{1-2D_2}V_{in2}. \tag{13}$$

With attention to the semiconductor devices at the output port (the secondary side of the DHSIC), an associated determination is discussed in the following. While all active switches are in OFF-state, the blocking voltages of diodes $D_5$ and $D_6$ are obtained by

$$V_{D5,stress} = V_{C3} + n_1(V_{C1} - V_{in1}) \tag{14}$$

and

$$V_{D6,stress} = V_{C4} + n_2(V_{C2} - V_{in2}), \tag{15}$$

respectively. The $V_{C1}$ and $V_{C3}$ can be founded by Equations (4) and (5) in turn, and $V_{C2}$ and $V_{C4}$ by Equations (8) and (9). As a result, the above Equations (14) and (15) can be rewritten as

$$V_{D5,stress} = \frac{2n_1}{1-2D_1}V_{in1} \tag{16}$$

and

$$V_{D6,stress} = \frac{2n_2}{1 - 2D_2} V_{in2},\tag{17}$$

respectively. Voltage stress of the output diode $D_O$ is estimated at the state that all active switches are closed and thus it will be

$$V_{Do,stress} = V_{C3} + n_1(V_{C1} - V_{in1}) + V_{C4} + n_2(V_{C2} - V_{in2}).\tag{18}$$

Based on Equations (4), (5), (8) and (9), the expression of Equation (18) is rewritten as

$$V_{Do,stress} = \frac{2n_1}{1 - 2D_1} V_{in1} + \frac{2n_2}{1 - 2D_2} V_{in2}.\tag{19}$$

### 3.3. Magnetizing Inductance Design

The minimum currents of active switches $S_1$ and $S_2$, $i_{Lm1,min}$ and $i_{Lm2,min}$, can be expressed as

$$i_{Lm1,min} = I_{Lm1} - \frac{\Delta i_{Lm1}}{2},\tag{20}$$

and

$$i_{Lm2,min} = I_{Lm2} - \frac{\Delta i_{Lm2}}{2},\tag{21}$$

respectively. The $\Delta i_{Lm1}$ and $\Delta i_{Lm2}$ stand for the current change on $L_{m1}$ and $L_{m2}$ in turn, while $I_{Lm1}$ and $I_{Lm2}$ are the average currents of $L_{m1}$ and $L_{m2}$. Assume that the converter is lossless. Then, The $I_{Lm1}$ and $I_{Lm2}$ can be determined as follows:

$$I_{Lm1} = \frac{2n_1}{(1 - 2D_1)} I_o,\tag{22}$$

and

$$I_{Lm2} = \frac{2n_2}{(1 - 2D_2)} I_o,\tag{23}$$

in which the $I_o$ denotes output current. In addition, $\Delta i_{Lm1}$ and $\Delta i_{Lm2}$ can be calculated by

$$\Delta i_{Lm1} = \frac{V_{Lm1,on}}{L_{m1}} D_1 T_s\tag{24}$$

and

$$\Delta i_{Lm2} = \frac{V_{Lm2,on}}{L_{m2}} D_2 T_s.\tag{25}$$

At the boundary, $\Delta i_{Lm1}$ and $\Delta i_{Lm2}$ are equal to zero, that is,

$$\frac{2n_1 V_o}{(1 - 2D_1)R_o} = \frac{V_{Lm1.on}}{2L_{m1}} D_1 T_s\tag{26}$$

and

$$\frac{2n_2 V_o}{(1 - 2D_2)R_o} = \frac{V_{Lm2,on}}{2L_{m2}} D_2 T_s.\tag{27}$$

From Equations (26) and (27), the minimum magnetizing inductances for CCM operation have to meet the following inequality:

$$L_{m1} > \frac{(1 - D_1)D_1 R_o V_{in1}}{2n_1 f_s\left(\frac{2n_1}{1-2D_1} V_{in1} + \frac{2n_2}{1-2D_2} V_{in2}\right)}\tag{28}$$

and

$$L_{m2} > \frac{(1 - D_2)D_2 R_o V_{in2}}{2n_2 f_s\left(\frac{2n_1}{1-2D_1} V_{in1} + \frac{2n_2}{1-2D_2} V_{in2}\right)}.\tag{29}$$

Suppose that the turns ratio $n_1 = n_2 = n$, $V_{in2} = 2V_{in1} = 24$ V, switching frequency $f_s = 40$ kHz, and both switches have the same duty ratio denoted as $D$. In addition, the converter operates at boundary condition mode (BCM) at 25 W. Figure 10 illustrates the relationship between magnetizing inductance and duty ratio for $L_{m1}$ and $L_{m2}$.

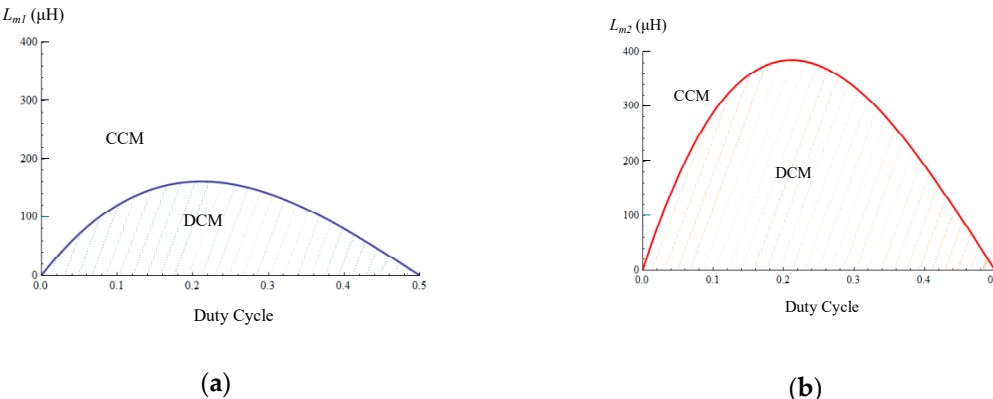

(**a**)                                                                                               (**b**)

**Figure 10.** The relationship between the magnetizing inductance and the duty cycle. (**a**) $L_{m1}$, (**b**) $L_{m2}$.

The comparison with other similar converters is summarized in Table 1. It can be found that even though the DHSIC needs more power components, it achieves excellent voltage gain over other DICs in addition to that the features of galvanic isolation, continuous input current and switch voltage clamping still can be possessed.

**Table 1.** Performance Comparison among the Proposed and Other Multi-Input Converters.

| Converter | Number of Switches | Number of Magnetic Components | Number of Capacitors | Input Current Ripple | Voltage Stress | Switch Voltage Camping | Galvanic Isolation | Output Voltage ($V_o$) |
|---|---|---|---|---|---|---|---|---|
| [15] | 2 | 2 | 5 | low | medium | no | no | $\frac{3}{1-D}V_{in1} = \frac{3}{1-D}V_{in2}$ |
| [16] | 4 | 2 | 2 | high | low | yes | yes | $n_1 D_1 V_{in1} + n_2 D_2 V_{in2}$ |
| [19] | 3 | 1 | 5 | high | medium | no | no | $\frac{3[D_1 V_{in1} + (D_1-D_2)V_{in2}]}{(1-D_3)}$ |
| [24] | 6 | 2 | 2 | low | high | no | no | $\frac{V_{in1}}{(1-D_1)} = \frac{V_{in2}}{(1-D_2)}$ |
| DHSIC | 4 | 2 | 5 | medium | low | yes | yes | $\frac{2n_1}{(1-2D_1)}V_{in1} + \frac{2n_2}{(1-2D_2)}V_{in2}$ |

## 4. Experimental Results

To verify the proposed DIC, a 200-W prototype is built, simulated and measured. Converter parameters and components adopted are summarized in Table 2. The control block diagram of the prototype is depicted in Figure 11. Power of input port 1 is calculated according to the detected input voltage and current, that is, $V_{in1,fb}$, and $I_{in1,fb}$, and then it is compared to a reference $P_{ref1}$. The control signals for $S_1$ and $S_2$ are determined by the PI power controller and the carrier. Accordingly, the input power at port 1 can be readily controlled. With respect to the other part of the control block diagram, the output voltage is regulated by controlling the switches $S_3$ and $S_4$. With such voltage regulation, the input port 2 can accommodate the supplement to output power and thus power dispatch at both input ports is accomplished. The proposed converter adopts MCU dsPIC30F4011 to serve as system controller. Figure 12 shows the control signals and the corresponding input currents at full load under DIO situation. In Figure 12 the duty ratios of $S_1$ and $S_2$ are 0.32 and 0.23, respectively, which is consistent with Equation (11) for a 400-V output. Meanwhile, the switch voltages, $v_{ds1}$ and $v_{ds2}$, are measured and shown in Figure 13, from which it can be observed that there is no voltage spike on active switches. That is, the boosting capacitors $C_1$ and $C_2$ are able to recycle leakage energy and can effectively clamp switch voltage. Additionally, withstood voltages on the switches at port 1 and port 2 are 34 V and 45 V, respectively, which verifies the theoretical analysis results of Equations (12) and (13). Figure 14 is the measurement of the voltages of boosting capacitors. This figure reveals that

voltages of $C_1$ and $C_2$ are in turn 34 V and 45 V, both voltage magnitudes of which meet the theoretical results of Equations (4) and (8). While step change takes place from half load to full load and then drops back, Figure 15 shows the related output voltage and current. Figure 15 illustrates that constant 400-V output still can be kept with even under step change. The overshoots at step-up and step-down transitions are only 1 V and 0.7 V, respectively. With respect to SIO situation, once power failure occurs at input 1, Figure 16 shows the measured control signal, output voltage, and input current. Similarly, if at input 2, Figure 17 is the related measurement. Both figures demonstrate the operation ability of the converter at SIO situation. The measured waveform of output current is presented in Figure 18, from which it can be observed that the output current of the converter is near to be ripple-free. The efficiency of the proposed converter is measured and then shown in Figure 19, in which the peak value is about 91.4%. In the experiment, a very low level of voltage, $V_{in1}$ = 12 V, is considered, therefore, which yields that higher current has to be demanded at a specific power, resulting in large conduction loss. This is the major reason why the converter efficiency is not as high as satisfied in the measurement. However, if input voltage is raised, the converter efficiency will be advanced. At single-input situation, if only the 24 V of $V_{in2}$ suppled the DHSIC, measured converter efficiency is shown in Figure 20. This figure reveals that a higher voltage input can yield a better efficiency even under the operation of single input. A photo of the built-up DHSIC is shown in Figure 21.

**Table 2.** Parameters of the DHSIC for Simulations and Practical Measurements.

| Parameter | Value |
| --- | --- |
| $V_{in1}$ (PV arrays) | 12–16 V |
| $V_{in2}$ (fuel-cells stack) | 24–30 V |
| $f_s$ (switching frequency) | 40 kHz |
| $V_o$ (output voltage) | 400 V |
| Power rating | 200 W |
| $L_{m1}$ and $L_{m2}$ (magnetizing inductors) | 176 μH and 302 μH |
| $L_{Lk1}$ and $L_{Lk2}$ (leakage inductors) | 1.9 μH and 2.4 μH |
| $S_1$–$S_4$ (power MOSFET) | IRFP4668 |
| $D_1, D_2, D_3,$ and $D_4$ | DSSK60-02AR |
| $D_5, D_6$ and $D_o$ | BYR29-600 |
| $C_1$ | 68 μF |
| $C_2$ | 33 μF |
| $C_3$ and $C_4$ | 22 μF |
| $C_o$ | 82 μF |
| $n_1$ and $n_2$ (transformer turns ratio) | 3 and 2.5 |

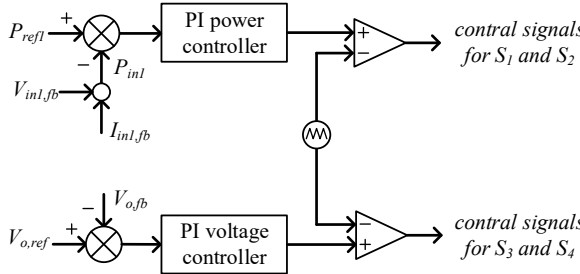

**Figure 11.** Control block diagram of the proposed converter.

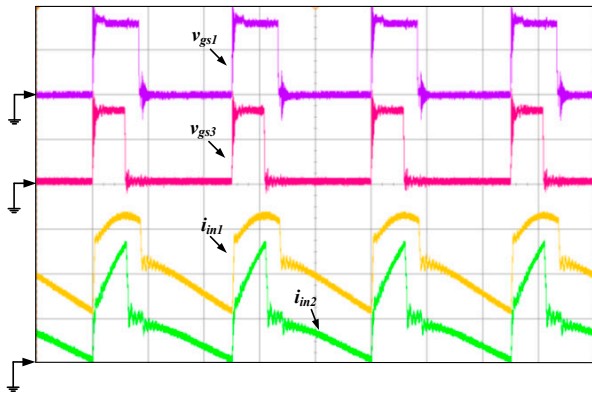

(control signals: 10 V/div, $i_{in1}$ *and* $i_{in2}$: 5 A/div, time: 10 μs/div)

**Figure 12.** The waveforms of control signals and corresponding input currents at full load under the DIO situation.

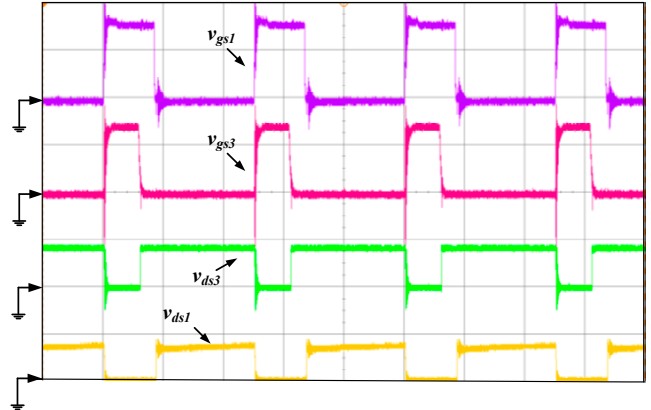

(control signal: 10 V/div, $v_{ds1}$ and $v_{ds2}$: 50 V/div, time: 10 μs/div)

**Figure 13.** The practical waveforms of the voltages across active switches.

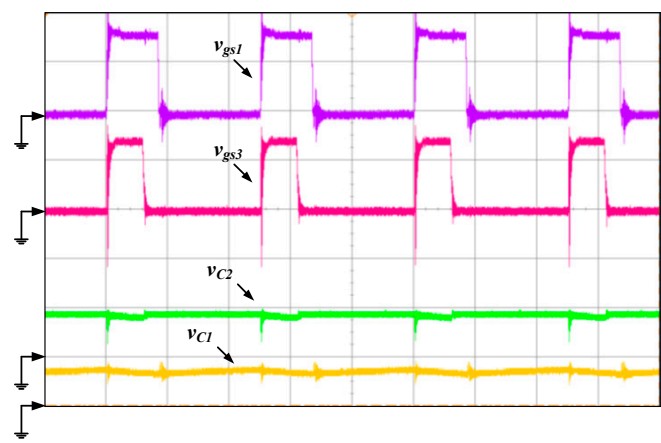

(control signals: 10 V/div, $v_{C1}$ and $v_{C\,2:}$ 50 V/div, time: 10 μs/div)

**Figure 14.** The voltage measurement of boosting capacitors $C_1$ and $C_2$.

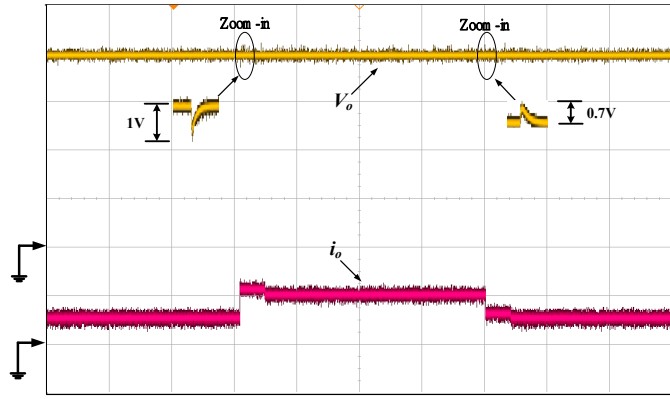

(*Vo*: 100 V/div, *io*: 0.5 A/div, time: 500 ms/div)

**Figure 15.** The waveforms of output voltage and current while step change occurs at load.

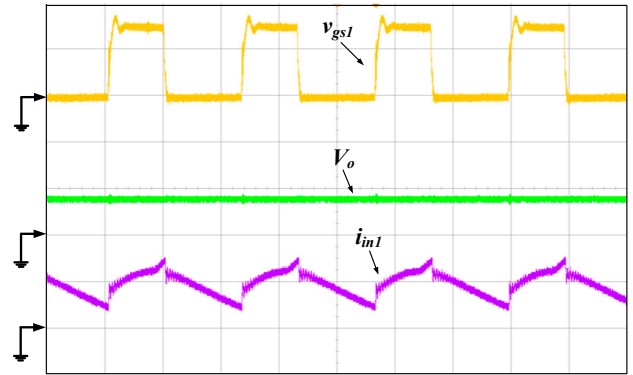

(*vgs1*: 10 V/div, *Vo*: 500 V/div, *iin1*: 10 A/div, time: 10 μs/div)

**Figure 16.** The measured waveforms of the control signal, output voltage, and input current while input 2 encounters power failure.

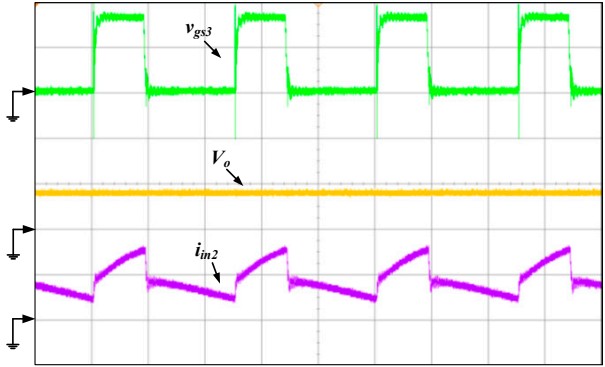

(*vgs3*: 10 V/div, *Vo*: 500 V/div, *iin2*: 10 A/div, time: 10 μs/div)

**Figure 17.** The measured waveforms of the control signal, output voltage, and input current while input 1 encounters power failure.

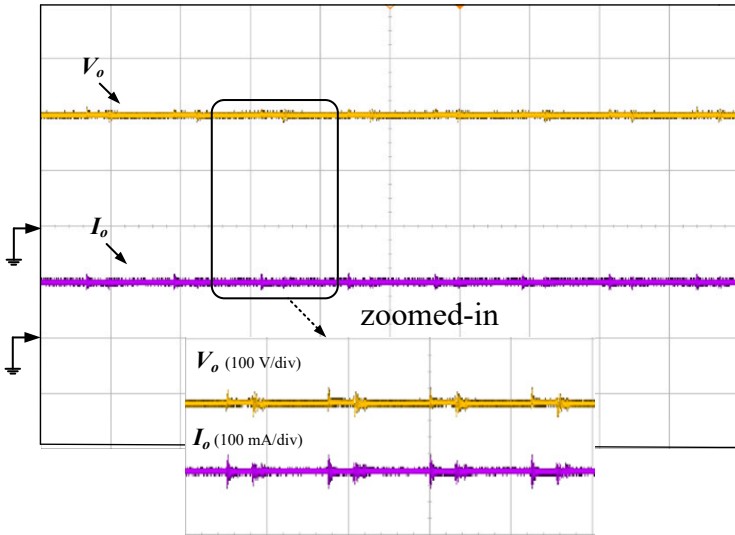

**Figure 18.** The practical measurement of output current $I_o$.

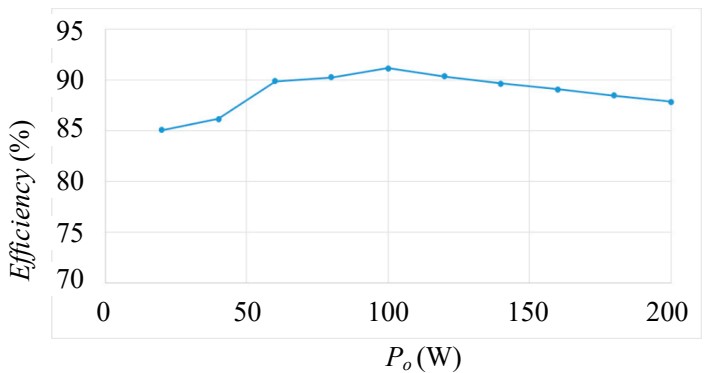

**Figure 19.** The conversion efficiency of the proposed converter.

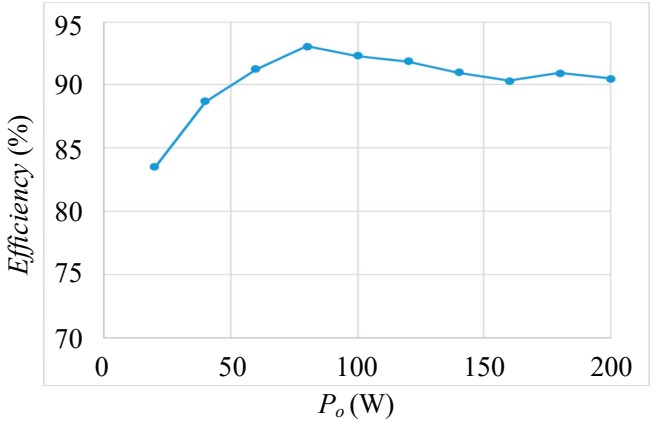

**Figure 20.** Measured converter efficiency while only input port 2 powers the DHSIC.

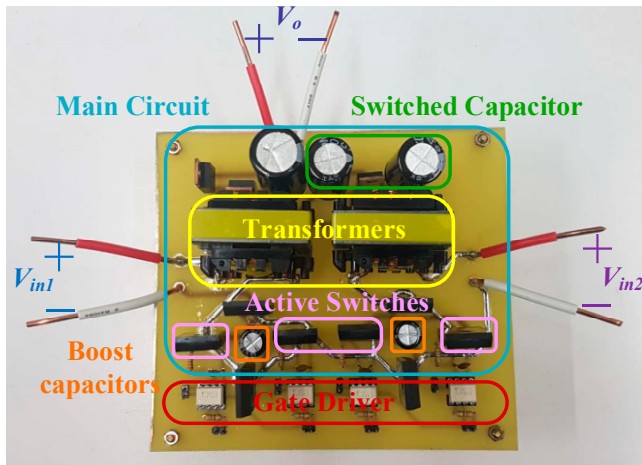

**Figure 21.** Photo of the prototype.

## 5. Conclusions

In this paper, a dual-input converter is proposed, which possesses the characteristics of ultra-high voltage gain, continuous input currents, galvanic isolation, inherent voltage-clamp feature, and recycling the energy stored in leakage inductance. This converter is capable of controlling the dual inputs independently and individually. Moreover, it still can accomplish all the mentioned features even under the case that either input encounters power failure. That is, the converter has operation flexibility to operate at dual-input situation or single-input situation for accommodating input conditions. A maximum of measured efficient is about 91.4% at dual-input operation.

**Author Contributions:** Conceptualization, C.-L.S., L.-Z.C. and H.-Y.C.; writing-original draft preparation, C.-L.S. and L.-Z.C.; writing-review and editing, C.-L.S. and L.-Z.C.; methodology, C.-L.S., L.-Z.C. and H.-Y.C.; validation, C.-L.S. and L.-Z.C.; supervision, C.-L.S.

**Funding:** This research received no external funding.

**Conflicts of Interest:** The authors declare no conflict of interest.

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
