# Peer review of "Dual-Input Isolated DC-DC Converter with Ultra-High Step-Up Ability Based on Sheppard Taylor Circuit"

_electronics, doi:10.3390/electronics8101125_

Round 1

Reviewer 1 Report

In this paper, the authors proposed a dual-input high step-up isolated converter (DHSIC). In order to improve the quality of the present manuscript, the following remarks should be addressed:

In this paper, the abstract is not concise. The authors need to very carefully communicative the contributions of their work in the context of other papers that have appeared in the literature. What are the limitations of the previous works as compared to the present manuscript, please explain it? The control implementation block diagram should be included in the revised manuscript and explained in more details in order that the reader can easily understand the presented concept of the implementation of the proposed converter. In order to make the paper more understandable to the reader, please explain both the simulation and experimental platforms in more details. In both simulation and experiment parts, the authors only presented the proposed method results. In order to show the superiority of the proposed scheme, the authors should compare the proposed method with these existing references. Finally, the English language should be improved in the entire paper.

Reviewer 2 Report

The paper is well presente and also experimental tests are carried out.
The proposed architecture is interesting
1) It is necessary to show a figure of the prorotype.
2) It is necessary to show a figure of the output current.
3) How the parameters of the converter are selected? Considering the high number of capacitor and inducactance,
the selection of the values is not very easy.
This is fundamental for a paper inherent the design of a new topologies.
4) Regarding the control: how the Dc-DC converter is controlled?
In particular, how it is possible to share the power between the two different input?

Reviewer 3 Report

1) Figure 1 is used to shows a hybrid system. It is clear that such dual source topology is required for this paper. However, its practical use is not justified. It is not usual to connect PV and fuel cell to the same inverter. A pv + battery system would be much more realistic. That being said, house systems with PV+battery do not use such inverter topologies. This aspect needs to be clarified. Reader should understand why Dual input is relevant.

2) DHSIC or DIHSC? needs harmonization

3) lie 90, why assume V2 is double of V1?

4) line 95, in state 1, figure 3a, why Itr1 and Ilm1 have different directions? if this is the initial state, starting state, shouldn't they have the same direction? Also, following dot conventions. if Itr1 and Itr2 flow into the dot, so should Id0 (current figure shows opposite directions for primary and secondary sides)

5) 193 is derived from Figure 7b not 7a.

6) Why figures 8 and 9 are only plotted for n = 1 to 3? any reason for not including higher turns ratio with higher gains?

7) for power electronics field, it is customary and much required to give a photo of the built up system. Section 4 requires a picture of the built inverter. Plots make sense, only after seeing the real hardware.

8) 91% inverter efficiency seems low. this needs to be discussed in section 4a and 5. It is good to see 1 input can still give same voltage boost, but the question of cost/benefit arises with such low efficiency

Round 2

Reviewer 1 Report

The authors did not response my previous round comments point to point. Please do that and also the reviewer wants to know how did the authors select the reference values in this paper. Please clarify this with a suitable explanation in the revised manuscript. 

Reviewer 2 Report

No comments.

Author Response

Thanks a lot.